# Country's Entrepreneurial Environment Predictors for Starting a New Venture—Evidence for Romania

**Carmen Păunescu** [1,*] and **Elisabeta Molnar** [2]

1   UNESCO Department for Business Administration, Bucharest University of Economic Studies,
    010374 Bucharest, Romania
2   Faculty of Economics and Business Administration, University of Szeged, H-6722 Szeged, Hungary;
    molnar.elisabeta@eco.u-szeged.hu
*   Correspondence: carmen.paunescu@ase.ro; Tel.: +40-747-776-700

**Abstract:** Entrepreneurship has been recognized as a key contributor to the economic development of countries and societal wellbeing. Building and sustaining an adequate entrepreneurial climate challenges—to a high extent—transitional economies world-wide, pushing these countries to develop policies and strategies aimed to sustain high-quality national entrepreneurship. The paper seeks to understand the key determinants of a country's entrepreneurial environment that drive potential entrepreneurs to assume an entrepreneurial status. It attempts to examine the countries' entrepreneurial environment factors that influence the development of entrepreneurial spirit and affect the potential entrepreneurs' decision to start a new venture as a desirable career opportunity. Entrepreneurial spirit is measured by entrepreneurial desirability, feasibility and social stability, taken from the Amway Global Entrepreneurship Report (AGER) 2018 data on national entrepreneurship. The results of the linear regression employed in the paper suggest that, in transitional economies like Romania, clear and stable rules and regulations, manageable taxes, an overall beneficial economic situation, as well as availability and accessibility of technology, may lead to greater entrepreneurial drive and ambition, which is fundamental to sustaining economic growth. The article ends with a discussion about the results and implications of the research.

**Keywords:** country's entrepreneurial environment; entrepreneurial spirit; starting a new venture; predictors; AGER; Romania

## 1. Introduction

Many studies on entrepreneurship acknowledge entrepreneurial activities as key drivers of growth, playing an important role in promoting economic and social development due to their capacity to aggregate job creation and productivity growth associated with the business entries that exceed exits [1–3]. Previous research concluded that conditions of the environment, including social, cultural, political and economic variables, are the main factor that affects one's aspirations to start a new venture [4], acknowledging the role of contextual barrier conditions in the development of intent to start a business [5,6]. Social, cultural and political barriers can affect entrepreneurship and, ultimately, affect people's willingness to participate in entrepreneurial activities. Research on entrepreneurship also pointed to the importance of individual traits, institutional factors and contextual characteristics in fostering the development of entrepreneurial intentions and actions [7–9]. Yet, Kallas (2019) [10] acknowledges that there are few approaches that take into account the external environment as a set of variables for making the decision to create an enterprise. She found that the perception of the potential entrepreneur of the political, economic, and socio-cultural environments shapes the future intention to start up a business. Even if the discussion of the environmental context in relation to

the entrepreneurial intention is not new, any study in the field is important as it contributes to the understanding of the entrepreneurship process. Thus, a good understanding of the environmental factors influencing the entrepreneurial process can largely contribute to the better equipping of the potential entrepreneurs with capabilities needed to leap from intention to entrepreneurship [11], as well as can help the policies aimed at supporting increased national entrepreneurship.

The entrepreneurship research on entrepreneurial potential and capabilities has pointed the importance that new ventures and start-ups hold for creating supportive and sustainable economic and societal wellbeing [11–15], and the incipiency level that entrepreneurial intention appears to have in transitional economies like Romania, despite the utmost potential that these countries intrinsically own [16,17]. On the whole, a transition economy is an economy that is changing from a centrally planned economy to a market-oriented economy. Since 2007, when Romania joined the European Union, the national economy has continued to undergo a set of structural transformations intended to develop entirely new free-market-based institutions. The establishment and growth of new enterprises are essential in the transition process. The development of the entrepreneurial sector and the success of entrepreneurship depend not only on initial conditions in the transition economies but also on the pace and firmness with which the reform process is run [18]. If properly assisted and supported through a clear and unshifting legislative basis, a supportive education system, as well as ongoing and stable financing sources, entrepreneurs could be able to produce powerful, long-lasting, and sustainable change for the benefit of their nations, regardless of economic challenges or social setbacks that they are confronted with [14,19]. Yet, earlier studies on entrepreneurial intentions and entrepreneurship career pathway development show that there is a decreasing interest in entrepreneurship and self-employment in many transitional economies around the globe [20–22].

In Romania, since 1990 in the post-communism landscape, the emergence of the legal framework for the business organization, as well as reorganization of work in the sense of acceptance of private forms of enterprises have generated a wave of entrepreneurs who have shaped a starved market for all kinds of products or services. However, the unstable economic climate, the lack of predictability and absence of infrastructure, the scarce job market and support services have generated tumultuous dynamics for small commercial businesses [17,23]. This has headed business owners to be dependent on business assistance through private networks.

An important aspect that is critical in attaining welfare is the employment status held by a person. The individuals not possessing a workplace and other categories that do not participate in the labor market have a higher exposure to poverty. Research by Callens and Croux (2009) [24] proved that the local context particularities can significantly influence the degree of poverty undertaken by people. Entrepreneurship is one of the most effective ways to generate employment and income. In Romania, the unemployment rate reached the lowest rate at the end of 2019 since 1991 in post-communism, however, youth unemployment remains a concern for policymakers. The economic recession or worse labor market conditions will push people into entrepreneurship out of necessity if they cannot get a job. However, people have the freedom to decide which type of work they want to do, if it is best for them to work for someone else or for themselves. The composition of the self-employed population reveals a greater gender gap in Romania than for the EU overall in 2019, with 26.2% self-employed women, a rate below the EU average (32.6%) [25]. More than 10% of the self-employed were under 30 years old and only 36.6% were above 50 years old [25]. Studies report that entrepreneurship in Romania is a career choice for all age groups, with a higher number of entrepreneurial activities than the European Union average recorded between 2013 and 2017 [26], and a proportion of the population that is involved in starting or managing a business, that is less than 42 months old, above the EU average for the period 2013–2017 (9.2% vs. 7.3%). Data from the European Commission (2017) [27] show that the working-age population will seemingly encounter a large projected decline from 67.1% in 2016 to 55.3% in 2070. Additionally, the young individuals and the segment of the population that has employment capabilities is steadily declining and is forecasted to encounter the same pattern in the future, making Romania vulnerable to social instability of any nature.

In 2018, in Romania, 135,532 new businesses were registered, while 80,181 deregistered [28]. Business entry rates have exceeded exit rates since 2013, being at about 10% for the past three years, which was above the median for EU Member States [28]. A large share of the workforce chose to develop entrepreneurial activity through incorporating as small businesses or acting as authorized licensed individual professionals, the simplest form of entrepreneurial activity organization in Romania [25]. These organizational forms were favored more by youth, who are generally more open to these flexible, atypical work arrangements [23,25]. The regulatory environment for start-ups in Romania is generally assessed as less favorable than the EU average. For example, the administrative burden on new start-ups ranks among the least favorable EU Member States [29]. In recent years, Romania improved the availability of services related to tax payments and tax regulations due to the modernization and partial digitalization of its tax collection system. Still, work needs to be carried out to facilitate the entrepreneurs' relationship with public authorities by using technology [26]. On the other hand, there is quite a wide array of financial support for business creation available in Romania. Since 2017, new tax incentives were designed to support enterprise development. Additionally, financial incentives, such as reduced social security contributions, are available for business creation by unemployed people [25,26].

Entrepreneurship can generate employment, innovation, and productivity increase. Reports by the European Commission (2019) [30] highlight some incongruities taking place within the European Union with respect to entrepreneurship development and innovation performance. Most of the Central and Eastern European countries appear to have a moderate degree of innovativeness, Romania being the exception since it is classified as a modest innovator and it stays below the European Union's average standing.

The current paper attempts to better elucidate the impact of the country's entrepreneurial environment factors on the decision to start a new venture by potential entrepreneurs. It also aims to broaden our understanding of the role of these predictor factors on entrepreneurial drive increase. Understanding the country context factors that affect the intention to start a new business venture allows also for improving policies aimed at sustaining these components of the entrepreneurial process [16]. As past literature has largely focused on unveiling the influencing factors of entrepreneurial intentions that pertain to and can be controlled by potential entrepreneurs themselves [19,31–39], the current study enriches the existing entrepreneurial cognition literature by examining prevalent country climate factors that enable and lead to an increased entrepreneurial ambition and desire. In the current paper, we argue that the country's entrepreneurial environment factors identified in the literature, such as manageable taxes, clear and stable operating environment, supportive education system, available technology, and beneficial economic situation, are the main predictors that enact and nurture a nation's entrepreneurial spirit and potential. The main objective of the current research is thus to empirically investigate the impact that these entrepreneurial environment factors have on Romania's entrepreneurial spirit. To do so, the current research uses the Amway Global Entrepreneurship Report (AGER) 2018 data, yielding robust results with respect to our objective. The research tests the empirically identified factors via linear regression. The present study complements a previous work that discussed the individual factors that drive the desirability of entrepreneurship in Romania based on AGER 2017 data [16]. The present work adds research evidence on the country's environmental factors that determine entrepreneurial ambition.

The development of the concept of entrepreneurial spirit has been supported by several studies to date [40–42]. In this regard, the existing studies have aimed to elucidate the role of education in building a positive entrepreneurial attitude as a fundamental pillar of the development of entrepreneurial spirit and also in developing and implementing entrepreneurial activities that demonstrate entrepreneurial drive. Previous research emphasized the need for entrepreneurs to actively adapt to the new economic normality specific to their country, and make necessary efforts to promote innovation and the development of enterprises [43]. Other research proposed a classification of the factors influencing the formation and development of the entrepreneurial spirit of youth, arguing on the importance of socio-cultural and economic environments specific to each country [44]. Thus, each individual country's

perspective with respect to the role of education in shaping the nation's entrepreneurial environment conditions will contribute to a better understanding of the field [45]. To date, little attention has been paid to the identification and empirical testing of the most prevalent entrepreneurial environment predictors that drive the entrepreneurial spirit of a nation, which will enable the development of government programs that foster values related to high-quality entrepreneurship [40,46]. The present study endeavors to advance our current understanding of a country's entrepreneurial climate factors that predict entrepreneurial spirit increase.

However, further theoretical and empirical research on cross-countries' environment predictors of entrepreneurial spirit and potential is needed to help guide future research, as well as improve the consistency and relevance of national economic strategies aimed at enhancing entrepreneurship. To this end, the use of global entrepreneurship data (provided, for example, by Global Entrepreneurship Monitor—GEM or AGER) can help further our understanding of the existing relationships between entrepreneurial spirit and its determinants. As a practical implication, the findings of the current research can help inform future policies aimed at stimulating national entrepreneurship in the transitional economies in Europe by delineating the most significant environmental factors impacting the entrepreneurial spirit of individual or groups of countries.

## 2. Literature Review

*Entrepreneurial Environment Factors that Fuel the Entrepreneurial Spirit*

It can be acknowledged that entrepreneurship comes as a viable solution for the resolution of disturbing communities or national-level phenomena, as well as a driver of economic growth and of the formation of a sustainable society. Due to its positive aid provided towards the alleviation of social and economic issues, but also as a driver of transformative change and growth, entrepreneurship has gained wide interest from the research side and, as a consequence, its benefits and challenges are highly discussed in the specialty literature [47–50]. The manner in which income, abundance and welfare are disseminated to community members positively influences the evolution of people's health degrees, life span, educational standing and happiness level, assisting them in becoming more entrepreneurial and being highly involved in society's processes. [51].

Being entrepreneurial means having the qualities that are needed to succeed as an entrepreneur. Building an entrepreneurial spirit means capturing the thinking, ambition and drive of the entrepreneurs as well as the qualities that fuel the actions they initiate [40]. Entrepreneurial spirit is an attitude and a drive that encompasses the desire to start a new venture as a desirable career opportunity [16,52]. It develops in individuals who demonstrate a true passion for building something great for their community regardless of the resources they hold and they are willing to push themselves to the limits to achieve big goals [40,42]. The entrepreneurship research literature discusses the development of entrepreneurial spirit by referring to the theory of planned behavior [53–56]. Thus, entrepreneurial spirit incorporates and is measured by three dimensions [16,53]: the desirability of entrepreneurship measured by entrepreneurial attitude and willingness to start a new venture as a desirable career opportunity; feasibility of entrepreneurship measured by possession of the necessary skills and resources to start a new business venture; and social stability measured by stability against social pressure exercised by family and friends that might dissuade potential entrepreneurs from starting a new venture.

First and foremost, new business ventures and enterprises highly need a surrounding national context that is always ready to offer the required elements for their good functioning, or else their quest for making the country a better place in terms of life quality may either slow-down in efficiency or stop. Decisional processes in the economic sphere, in this sense, must start at the governmental level, but not all countries are able to be active supporters of the entrepreneurial movement and innovative change [57]. Entrepreneurship research evidence [58,59] indicates that there needs to be suitability between new ventures' assistance and the infrastructural elements, including legislative provisions

that come to the aid of these societal actors. Furthermore, Urban and Kujinga (2017) [60] debate on the powerful influence held by governmental institutions in setting the framework within which entrepreneurship is conducted and the agreeableness that entrepreneurial activities imply. Through their great influence, governmental entities are also capable of enhancing bureaucracy, for example, through strict procedures that must be followed and lack of funding availability. This practice could be very discouraging in the case when a potential entrepreneur wishes to act on an opportunity that he has identified in an emerging economy [48]. Lee et al. (2015) [8] found that government support had a negative impact on the number of new businesses in the countries where the development of financial support, college education, and opportunities for entrepreneurship is well advanced. Entrepreneurs should focus not only on developing their competitive resources but also on matching them with the context created by the formal institutional settings.

The challenges faced by new ventures in the modern context seem to be numerous, Borza et al. (2009) [61] pinpointing the non-existence of governmental support, the absence of regulations and the poor recognition of regional development prospects. Apart from the aforementioned constraints, new ventures are also limited by the identification of adequate financing methods, which help them to fuel the carrying of their mission. Previous research [62] pointed out the role of governments in improving access to capital by utilizing public financial resources, lowering barriers to entry for new business ventures, and developing programs to support entrepreneurs to grow beyond the initial start-up phase. However, there is insufficient knowledge on the degree to which the supportive procedures implemented by governmental institutions will manage to make entrepreneurial intentions to be intensified [63]. Therefore, our first hypothesis states that:

**Hypothesis 1 (H1).** *The clearer is the country's operating environment, the stronger is the individuals' entrepreneurial spirit.*

Even if governmental institutions constitute a major authoritative force in establishing the degree to which entrepreneurial actions are carried out, literature notes other aspects that can either diminish or boost the success of such intentions, such as education. Education can be considered one of the most valuable pillars of human development as it provides the necessary chances of accessing a solid, sustainable future existence, and in the view of Datzberger (2018) [64], it constitutes an essential human entitlement. Tarabini and Jacovkis (2012) [65] refer to the stringent need of reshaping the educational system in such a way that poor individuals can be granted privileges and offered help towards increasing their opportunities and strengthening their income-producing abilities, potentially through entrepreneurship and self-employment. The study of Hidalgo-Hidalgo and Iturbe-Ormaetxe (2018) [66] raises concern over this matter and finds out that the expenses undertaken at the public level with respect to education can indeed display promising results in terms of improving life opportunities on the long term. Maloma (2016) [67] considers that the absence of education can make the individual who is affected by poverty minimize earning possibilities, therefore, in line with previous and subsequent research [68–70], he indicates that the undertaking of investments in education represents an essential action step in fighting against poverty, as it enhances one's self-employment opportunities and as such, boosts prosperity.

Moreover, the commitment of the entrepreneurs [71], their skills and competence in noting previously un-tackled gaps and their risk approach, their leadership abilities, expertise, educational background and competencies with respect to innovation [72], and the involvement in entrepreneurial incubators and support hubs through the help of academic institutions [73,74] can also boost the success of entrepreneurial drive and potential. Based on these arguments, our second hypothesis is formulated as follows:

**Hypothesis 2 (H2).** *The more supportive is the national education system, the stronger is the individuals' entrepreneurial spirit.*

Despite the fact that new ventures require skills, knowledge and certain personality attributes to be run, it appears that the costs that are needed do not manage to exceed the benefits that these entities create for the society [75]. A good economic and social standing of a country comes with a cost that can be seen firstly in increasing welfare expenditure, since in these countries, the government acts as a provider of socially directed services (health, education, caring for the elderly) by relying on the taxes that it collects [76,77]. Knowing the fact that these services are labor-dependent, an increase in their productivity level will trigger the need for higher wages and people will have to pay higher taxes to support the increase in social services' costs. The incremental change in taxes will be reflected in discouragement of the entrepreneurial field and, consequently, entrepreneurship will be at risk. Therefore, our third hypothesis states:

**Hypothesis 3 (H3).** *The more manageable are the taxes, the stronger is the entrepreneurial spirit.*

The spread of technology gives rise to new methods of allowing enterprises to design, develop, package, market, and distribute their products in the global marketplace. Beliaeva et al. (2020) [78] states that digital technologies have nowadays a significant impact on how new business ventures are imagined and created. The arising technology paradigm is leveraging the potential of collaboration and collective intelligence to design and launch more robust and sustainable entrepreneurial projects [79,80]. Technological changes prompt countries to employ investments in human capital, since the process of digitalization operates modifications to the employment domain, the requirements towards people's capabilities and ultimately to the social defense structures. As a consequence, nations must pay attention to the possibility of these technological improvements to avoid bringing inequity and social exclusion to communities [81].

Not all jobs will require a high level of tech skills, but most jobs will require basic tech skills, such as data analysis or data literacy [82]. In the last ten years, technology trends such as mobile services, social media, cloud computing, Internet of Things (IoT), big data and robotics [80] have supported new ways of collaborating, organizing resources, designing products, matching complex demand and offer, and developing new standards and solutions [79]. Such rapid development has profoundly changed the competitive environment and reshaped traditional business strategies, models and processes [78]. Thus, the biggest challenge for entrepreneurs in the digital era is not collecting data, but being able to comprehend them and analyze them, in order to empower their employees to make informed decisions, improve productivity in the workplace and gain a competitive advantage on the market. Zaheer et al. (2018) [83] confirm that a transformation of existing economies into digital represents a significant opportunity for the accelerated growth of the local and national economy, whose drivers are small and medium-sized enterprises. According to Yin and Du (2018) [84], enterprises, including new ventures and start-ups, that fail to adapt adequately to the digital environment will inevitably face a decline in competitiveness, and some of them may face their very survival on the market. Unlike in the past years, it is no longer enough to produce cheaper, faster, and high-quality goods [84]. Enterprises are forced to come up with new forms of innovative digital technologies that can sustain the current competitive advantage in the long-term. Entrepreneurs are expected to demonstrate the ability to work across the environmental boundaries and ensure the transfer and exchange of knowledge in society, through collaboration with research universities, and the development of incubators and business portals [79]. Based on these arguments, our fourth hypothesis is formulated as follows:

**Hypothesis 4 (H4).** *Technology availability positively relates to an increased entrepreneurial spirit and makes entrepreneurship pathway easier.*

The study of van der Paauw (2016) [85] underlines that the choice of the business approach under a given economic context is decisive for how managerial practices are enacted, how the set objectives will be delivered and how external risks are fought against, ultimately shaping success or ending in failure. A positive or beneficial economic climate is considered when the national economy is

expanding. Ensuring that the population who live in this economic environment experience a high level of life quality is an essential aspect for providing people real chances of (self) development and self-employment. Beneficial economic conditions for them is associated to aspects such as: physical circumstances, economic, social and cultural considerations, the quality of their social relationships, the goods and services that they produce or reach, consumption patterns, their manner of living, their degree of fulfillment or non-fulfillment, and feelings such as joy or disappointment [86].

Institutional elements that influence the living standards of the population and the rate at which innovation is created and diffused are the legislative structure, financial matters, and the provision of knowledge and information [87]. In the assessment of the economic conditions and determination of economic progress, it is also noteworthy to consider the density of a territory's population, the stage of economic development in terms of income per capita and the number of individuals that possess entrepreneurial educational attainment, to grasp the degree of innovativeness at the local and regional level [88,89]. Based on these arguments, our last hypothesis is formulated as follows:

**Hypothesis 5 (H5).** *A beneficial economic situation in the country positively relates to an increased entrepreneurial spirit.*

## 3. Materials and Methods

The current paper seeks to understand the key determinants of a country's entrepreneurial environment that drive individuals to assume an entrepreneurial status. It attempts to examine a country's entrepreneurial environment conditions that influence the decision to start a new business venture as a desirable career opportunity. The paper's objectives are three-fold: (1) to identify the driving factors of a country's entrepreneurial environment that predict the individual's entrepreneurial spirit; (2) to understand the impact of entrepreneurial environment predictors on entrepreneurial drive and ambition in Romania; and (3) to determine whether a particular factor (variable) is still able to predict the entrepreneurial spirit when the effects of another factor (variable) are controlled for.

The paper uses the Amway Global Entrepreneurship Report (AGER) 2018 dataset for Romania. Since 2011 globally and 2013 in Romania, AGER has examined different characteristics of entrepreneurship in order to make it more attractive to potential entrepreneurs and stimulate interest in pursuing a self-employment career. Each year, the study was carried out globally by the prestigious marketing research institute GfK Consumer Experiences, Nuremberg, Germany, and it covers 44 nations in the 2018 AGER edition. The study was carried out within the framework of an omnibus survey with the data collected face-to-face and validated through telephone interviews. The selection procedures used ensured a representative sample in each participating country. The data were collected using a fully structured questionnaire, with the interviewers having to follow questions in a given order and wording. The omnibus survey used a stratified sample, a probabilistic sampling technique, with the population divided into subgroups who all shared a similar characteristic. The period of fieldwork lasted from 21 April 2017 to 12 May 2017. The sample size of Romanian population analysed in the 2018 edition of AGER was 1038, and included: youngsters (n = 219), seniors of 60 + years (n = 298), females (n = 531), males (n = 507), low income population with a monthly household net income lower than 1000 RON (n = 155), university degree (n = 191), and working persons (n = 511).

The variables tested in the research, indicated by the specialty literature as predicting factors of the country's entrepreneurial environment that affect an individual's entrepreneurial spirit, are: Taxes (Tax)—measured by the extent to which dealing with taxes in a country is perceived as being manageable [76,77]; Operating environment (OE)—measured by the extent to which the existing rules and regulations of a country are perceived as being easy to understand and follow [59,60]; Education system (EduS)—measured by the extent to which people perceive that their national education system is supportive and helps them develop the entrepreneurial skills that they need [68,73]; Technology (Tech)—measured by the extent to which technology is perceived as being available and accessible to make entrepreneurship pathway easier [83,84]; Economic situation (EcS)—measured by the extent

to which the economic situation in the country is perceived as being positive and beneficial [88,89]. According to AGER, the Entrepreneurial spirit (EntrepS) is measured by the mean score of the individuals' perceived entrepreneurial desirability, feasibility and social stability [53,54].

The research is exploratory and employs a linear regression model with control variables. Multiple linear regression is a predictive analysis used to explain the relationship between one continuous dependent variable and two or more independent variables [90]. It estimates how much the changes in each predictor variable relate to changes in the outcome variable. Multiple regression is also employed to measure whether including an additional variable makes a difference. Thus, it is used to control for other variables when exploring the predictive ability of the model. Controlling variables is important because slight variations in the predictor variables could strongly affect the outcome being measured. We use linear regression to understand whether the entrepreneurial spirit can be predicted based on a country's entrepreneurial environment factors, such as: manageable taxes, clear rules and regulations, supportive education system, available technology, and beneficial economic situation, as well as based on demographic factors such as: age, gender, education, and working status.

## 4. Results

### 4.1. Key Figures regarding Entrepreneurial Environment Factors in Romania

According to AGER 2018, an analysis of the entrepreneurial environment driving factors for starting and running a new venture in Romania, as these were perceived by respondents, shows that people acknowledge first and foremost the role of a supportive education system on preparing individuals for a potential entrepreneurial career path. Afterward, it follows the importance of availability and accessibility of technology, the necessity of dealing with manageable taxes and the clarity and stability of the operating environment (Figure 1). By contrast, the average EU 28 and global average countries give prevalence to technology availability first, then a supportive education system and a beneficial economic situation in these countries.

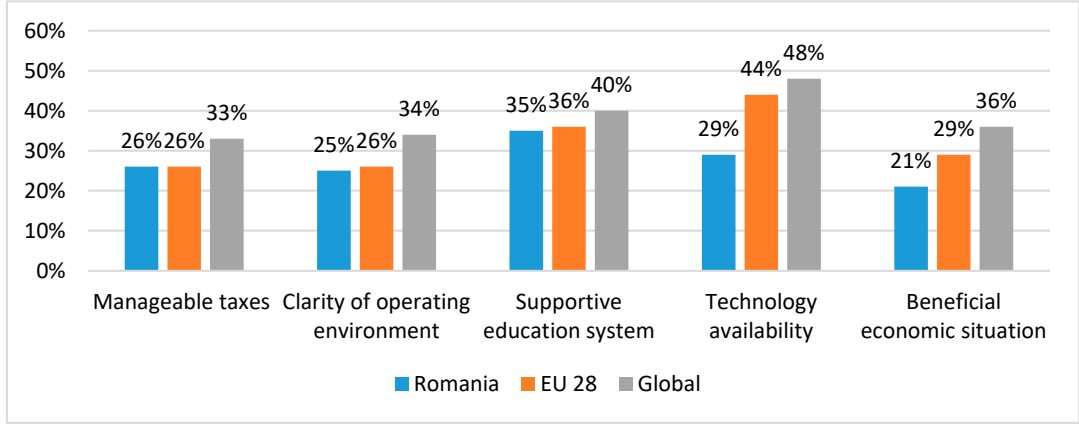

**Figure 1.** Entrepreneurial environment factors in Romania vs. EU 28 and the global level. Source: Authors' computation based on Amway Global Entrepreneurship Report (AGER) 2018 data.

Looking at the differences and similarities by gender and age groups, the trend lines follow the same pattern for both men and women (Figure 2), with a supportive education system coming first, then technology availability and manageable taxes. Romanians under 35 years and, surprisingly, people over 50 years emphasize the importance of a supportive education system and technological availability in the first place, to different extents (Figure 3).

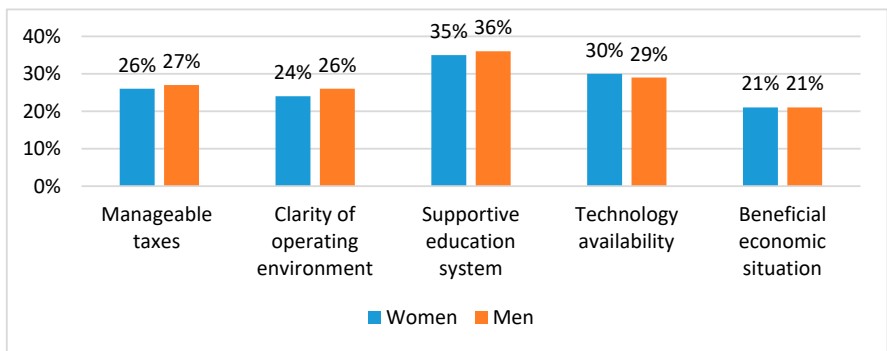

**Figure 2.** Entrepreneurial environment factors for women vs. men in Romania. Source: Authors' computation based on AGER 2018 data.

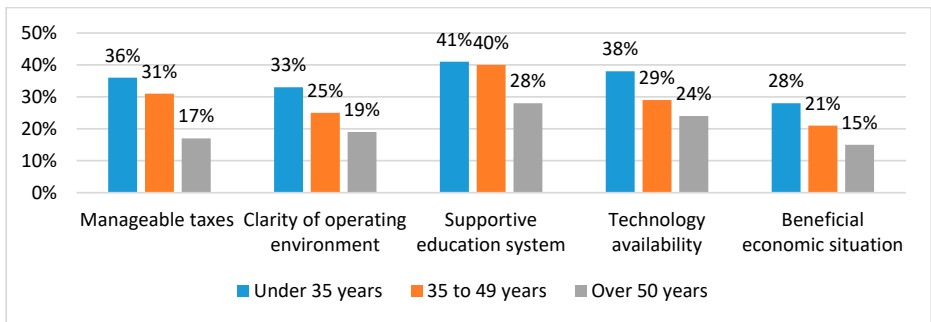

**Figure 3.** Entrepreneurial environment factors for different age groups in Romania. Source: Authors' computation based on AGER 2018 data.

*4.2. Regression Analysis of Romania's Entrepreneurial Environment Factors*

The following section provides a concise description of regression analysis and interpretation of the results.

Multiple linear regression is conducted, by using the Enter method of regression, to understand whether the entrepreneurial spirit (dependent variable) can be predicted based on the variables which describe a country's entrepreneurial environment (independent variables), namely: taxes, rules and regulations, education system, technology, and economic situation. We explored the causal relationship between the variables tested and their predictive power while statistically controlling for another variable. We suspected that the relationship between variables that describe a country's entrepreneurial climate may be influenced by the impact of other variables. The variables normally controlled for are socio-demographic variables such as age, gender, education, and working status. The regression model was run to understand the relative contribution of each of the predictors, namely socio-demographic variables, respectively country's entrepreneurial climate factors, to the total variance explained. Pearson's correlations (Table 1) indicate that all the variables included in the study are significantly moderate or strongly correlated, the p-level of significance being lower than 0.01 or 0.05. Correlations with the dependent variable (entrepreneurial spirit) are higher for age-groups ($r = -0.372$), taxes ($r = 0.290$), working status ($r = -0.257$), economic situation ($r = 0.255$) and operating environment ($r = 0.236$), and lower in case of technology ($r = 0.213$) and education ($r = 0.205$). Looking at the mean scores, the closer the mean scores to the middle point (0.5), the higher their corresponding variables account for an increased entrepreneurial spirit. A supportive education system (xbar = 0.35, SD = 0.477) seems to contribute, to a higher extent, to the development of entrepreneurial spirit. The mean scores for socio-demographic variables indicate that there is a balanced number of respondents from each age-group, each gender, average respondents completed at least secondary or high school, and on average they have part-time jobs. The values of standard deviation in the statistical dataset close to the mean convey that these variables are highly reliable. All the variables entered the model.

**Table 1.** Mean, standard deviation and Pearson's correlations.

| | | Mean | SD | Age | Sex | Edu | WS | Tax | OE | EduS | Tech | EcS | EntrepS |
|---|---|---|---|---|---|---|---|---|---|---|---|---|---|
| Age (in groups) | Pearson Correlation | 4.16 | 1.765 | 1 | 0.007 | −0.197 ** | 0.365 ** | −0.181 ** | −0.130 ** | −0.121 ** | −0.113 ** | −0.114 ** | −0.372 ** |
| Sex | Pearson Correlation | 1.51 | 0.500 | | 1 | −0.073 ** | 0.108 ** | −0.008 | −0.019 | −0.009 | 0.002 | 0.004 | −0.053 * |
| Education (Edu) | Pearson Correlation | 2.97 | 0.692 | | | 1 | −0.396 ** | 0.095 ** | 0.042 * | −0.012 | 0.013 | 0.014 | 0.205 ** |
| Working status (WS) | Pearson Correlation | 2.54 | 1.478 | | | | 1 | −0.090 ** | −0.031 | −0.057 * | −0.029 | −0.084 * | −0.257 ** |
| Taxes (Tax) | Pearson Correlation | 0.26 | 0.440 | | | | | 1 | 0.341 ** | 0.258 ** | 0.293 ** | 0.378 ** | 0.290 ** |
| Operating environment (OE) | Pearson Correlation | 0.25 | 0.432 | | | | | | 1 | 0.354 ** | 0.387 ** | 0.393 ** | 0.236 ** |
| Education system (EduS) | Pearson Correlation | 0.35 | 0.477 | | | | | | | 1 | 0.388 ** | 0.295 ** | 0.186 ** |
| Technology (Tech) | Pearson Correlation | 0.29 | 0.456 | | | | | | | | 1 | 0.315 ** | 0.213 ** |
| Economic situation (EcS) | Pearson Correlation | 0.21 | 0.406 | | | | | | | | | 1 | 0.255 ** |
| Entrepreneurial spirit (EntrepS) | Pearson Correlation | 0.28 | 0.330 | | | | | | | | | | 1 |

** Correlation is significant at the 0.01 level (2-tailed); * Correlation is significant at the 0.05 level (2-tailed). Source: Authors' computation based on AGER 2018 data.

The regression model summary is presented in Table 2. There are two models listed. Model 1 refers to the control variables: age (in groups), gender, education, and working status, while Model 2 includes all the predicting variables of the entrepreneurial spirit: age, gender, education, working status, taxes, operating environment, technology, economic situation, and education system. Checking the R square value in the third column, Model 1 explains 16.5% of the variance, while Model 2 that includes all predictor variables (including socio-demographic variables) explains 24.5% of the overall variance in the entrepreneurial spirit. To establish how much of this overall variance is explained by our variables of interest after the effects of age, gender, education and working status are removed, we look at the column labeled R square change ($\Delta R^2$). The overall variance explained by the R square change after the removal of control variables explains the additional 8.0% significant amount of variance. This means that technology, economic situation, education system, taxes and operating environment explain an additional 8.0% increase of the entrepreneurial spirit. This is a statistically significant contribution, as indicated by the Sig. F Change value for this line (0.001).

**Table 2.** Regression model summary [c].

| Model | R | $R^2$ | Adjusted $R^2$ | Std. Error | Change Statistics | | | | |
|-------|---|-------|----------------|------------|-------------------|---|---|---|---|
| | | | | | $\Delta R^2$ | $\Delta F$ | df1 | df2 | Sig. F Change |
| 1 | 0.407 [a] | 0.165 | 0.162 | 0.30285 | 0.165 | 51.165 | 4 | 1033 | 0.000 |
| 2 | 0.495 [b] | 0.245 | 0.239 | 0.28809 | 0.080 | 21.805 | 5 | 1028 | 0.000 |

[a] Predictors: (Constant), WS, Sex, Age, Edu; [b] Predictors: (Constant), WS, Sex, Age, Edu, Tech, EcS, EduS, Tax, OE; [c] Dependent Variable: Entrepreneurial spirit. Source: Authors' computation based on AGER 2018 data.

The ANOVA results (Table 3) show that all our variables are significant predictors of the entrepreneurial spirit and, therefore, of the intention to start a new business venture (Model 2: F = 37.143, $p < 0.001$), and that the model is a good fit.

**Table 3.** Analysis of Variance (ANOVA) [a].

| | Model | Sum of Squares | df | Mean Square | F | Sig. |
|---|-------|----------------|-----|-------------|---|------|
| | Regression | 18.697 | 4 | 4.674 | 51.165 | 0.000 [b] |
| 1 | Residual | 94.371 | 1033 | 0.091 | | |
| | Total | 113.068 | 1037 | | | |
| | Regression | 27.746 | 9 | 3.083 | 37.143 | 0.000 [c] |
| 2 | Residual | 85.323 | 1028 | 0.083 | | |
| | Total | 113.068 | 1037 | | | |

[a] Dependent Variable: Entrepreneurial spirit; [b] Predictors: (Constant), WS, Sex, Age, Edu; [c] Predictors: (Constant), WS, Sex, Age, Edu, Tech, EcS, EduS, Tax, OE. Source: Authors' computation based on AGER 2018 data.

To find out how well each of the variables predicts the dependent variable (entrepreneurial spirit), we look at the coefficients (Table 4). The coefficients (Model 2) show that manageable taxes (b = 0.099, $p < 0.001$), beneficial economic situation (b = 0.088, $p < 0.001$), clear and stable operating environment (b = 0.053, $p < 0.05$), and technology availability (b = 0.049, $p < 0.05$) are significant predictors of the development of entrepreneurial spirit. Based on our data, the education system seems not to be a significant predictor of the entrepreneurial drive to start a business. Additionally, age, education level and working status are significant predictors of entrepreneurial spirit. As the model of regression shows, a 1-unit increase in "manageable taxes" will result in a 0.099 unit increase in the entrepreneurial spirit. Thus, the easier is dealing with taxes, the stronger is entrepreneurial drive. Additionally, a 1-unit increase in "beneficial economic situation" will result in a 0.088 unit increase in the entrepreneurial drive for a new business venture and a 1-unit increase in "easy to understand and follow rules and regulations" will result in a 0.053 unit increase in the entrepreneurial spirit. This means that the better the economic situation is and the clearer the rules and regulations are,

the stronger the entrepreneurial drive is. Finally, 1-unit increase in technology availability leads to a 0.049 unit increase in the entrepreneurial spirit. As such, the research hypotheses H1, H3, H4 and H5 are successfully confirmed.

**Table 4.** Coefficients [a].

| Model | | Unstand. Coeff. | | Stand. Coeff. | t | Sig. | 95.0% Confidence Interval for b | | Correlations | | | Collinearity Statistics | |
|---|---|---|---|---|---|---|---|---|---|---|---|---|---|
| | | b | Std. Error | Beta | | | Lower Bound | Upper Bound | Zero-order | Partial | Part | Tolerance | VIF |
| 1 | (Constant) | 0.475 | .065 | | 7.332 | 0.000 | 0.348 | 0.603 | | | | | |
| | Age | −0.059 | 0.006 | −0.316 | −10.338 | 0.000 | −0.070 | −0.048 | −0.372 | −0.306 | −0.294 | 0.862 | 1.159 |
| | Sex | −0.022 | 0.019 | −0.033 | −1.154 | 0.249 | −0.059 | 0.015 | −0.053 | −0.036 | −0.033 | 0.986 | 1.014 |
| | Edu | 0.048 | 0.015 | 0.101 | 3.265 | 0.001 | 0.019 | 0.077 | 0.205 | 0.101 | 0.093 | 0.839 | 1.191 |
| | WS | −0.022 | 0.007 | −0.098 | −2.981 | 0.003 | −0.036 | −0.007 | −0.257 | −0.092 | −0.085 | 0.751 | 1.331 |
| 2 | (Constant) | 0.356 | 0.063 | | 5.632 | 0.000 | 0.232 | 0.480 | | | | | |
| | Age | −0.049 | 0.006 | −0.263 | −8.845 | 0.000 | −0.060 | −0.038 | −0.372 | −0.266 | −0.240 | 0.833 | 1.201 |
| | Sex | −0.021 | 0.018 | −0.032 | −1.183 | 0.237 | −0.057 | 0.014 | −0.053 | −0.037 | −0.032 | 0.985 | 1.015 |
| | Edu | 0.046 | 0.014 | 0.097 | 3.259 | 0.001 | 0.018 | 0.074 | 0.205 | 0.101 | 0.088 | 0.831 | 1.203 |
| | WS | −0.021 | 0.007 | −0.092 | −2.941 | 0.003 | −0.034 | −0.007 | −0.257 | −0.091 | −0.080 | 0.746 | 1.340 |
| | Tax | 0.099 | 0.023 | 0.131 | 4.273 | 0.000 | 0.053 | 0.144 | 0.290 | 0.132 | 0.116 | 0.776 | 1.288 |
| | OE | 0.053 | 0.024 | 0.069 | 2.153 | 0.032 | 0.005 | 0.100 | 0.236 | 0.067 | 0.058 | 0.718 | 1.393 |
| | EdS | 0.023 | 0.021 | 0.033 | 1.072 | 0.284 | −0.019 | 0.064 | 0.186 | 0.033 | 0.029 | 0.777 | 1.286 |
| | Teh | 0.049 | 0.023 | 0.067 | 2.156 | 0.031 | 0.004 | 0.093 | 0.213 | 0.067 | 0.058 | 0.750 | 1.333 |
| | EcS | 0.088 | 0.025 | 0.108 | 3.463 | 0.001 | 0.038 | 0.138 | 0.255 | 0.107 | 0.094 | 0.748 | 1.337 |

[a] Dependent Variable: Entrepreneurial spirit. Source: Authors' computation based on AGER 2018 data.

The standardized beta values (Model 2) indicate that manageable taxes ($\beta = 0.131$, $t(1038) = 4.273$) and beneficial economic situation ($\beta = 0.108$, $t(1038) = 3.463$) have the most impact on the perceived entrepreneurial spirit and drive, followed by clear operating environment ($\beta = 0.069$, $t(1038) = 2.153$) and availability of technology ($\beta = 0.067$, $t(1038) = 2.156$). The values of VIF (variance inflation factor) close to 1 indicate that there is no collinearity found between independent variables and, as such, in our regression model all predictor variables can independently predict the value of the dependent variable.

A supportive education system cannot predict an increased entrepreneurial spirit significantly, as the t-test for equality of means generated a t-value of 1.072 with a p significance level higher than the 0.05 threshold for significance. Therefore, Hypothesis H2 cannot be proved based on the existing data analyzed in the paper and is rejected.

The summary of our research results is shown in Table 5.

**Table 5.** Research results.

| Hypothesis | Description | Variable | Coefficient | Findings |
|---|---|---|---|---|
| H1: | The clearer is the country's operating environment, the stronger is the individuals' entrepreneurial spirit. | Operating environment | 0.053 | $p < 0.05$, supported |
| H2: | The more supportive is the national education system, the stronger is the individuals' entrepreneurial spirit. | Education system | 0.023 | $p > 0.05$, rejected |
| H3: | The more manageable are the taxes, the stronger is the entrepreneurial spirit. | Taxes | 0.099 | $p < 0.001$, supported |
| H4: | Technology availability positively relates to an increased entrepreneurial spirit and makes entrepreneurship pathway easier. | Technology | 0.049 | $p < 0.05$, supported |
| H5: | A beneficial economic situation in the country positively relates to an increased entrepreneurial spirit. | Economic situation | 0.088 | $p < 0.001$, supported |

## 5. Discussion and Conclusions

The investigation of the impact of entrepreneurial spirit on the economic growth of nations worldwide has been one of the main justifications of the AGER project. The present paper critically analyzed the most prevalent factors of a country's entrepreneurial environment, indicated by the

specialty literature, that impact the entrepreneurial spirit of potential entrepreneurs in Romania. The results show that the process of developing a new business venture can encounter many challenges. Challenges are typically external in nature (taxation, legal framework, technology availability, education system, economic situation) and, usually, are not under the influence of new ventures, but they can have a consistent impact on them.

Romania has the potential of becoming a knowledge society and of fostering sustainable advancement from the economic and social perspective by intensifying entrepreneurial activity and embedding knowledge and innovation at the institutional level [91]. However, the country is severely confronted across the nation with the lack of employment possibilities, demographic aging, and growing poverty rates [92]. Performing research to determine the capabilities that influence the implementation of entrepreneurial projects across the country, de los Ríos-Carmenado et al. (2014) claim that Romanians do not create efficient results because of human-linked considerations, for example, their beliefs and their conduct when it comes to self-employment and entrepreneurship [93]. Since human capital is a critical asset in generating a thriving society, the bottom-line priorities for a Romanian entrepreneurial society should be thus directed toward investments in educational infrastructure, digital entrepreneurship and increase of performance at the institutional level [91,92]. Therefore, Romania meets the premises that allow it to struggle for progress in sustainable terms.

The present study comes as a completing element of entrepreneurship cognition research in order to fill in the reduced number of scientific research studies on the environment-predicting factors of entrepreneurial spirit in Romania.

Through the study, it was confirmed through linear regression that the degree to which dealing with taxes in a country is perceived as being manageable and the degree to which the economic situation in the country is perceived as being beneficial for entrepreneurial activity are stronger predictors of the development of entrepreneurial spirit and drive of Romanians. These findings are consistent with previous works that found that the intensity of entrepreneurial activity is dependent upon the stage of economic development of the country [77] and economic competitive growth generated by entrepreneurial activity is impacted by taxation and investments in innovation and development of labor resources [89].

The paper results also showed that clarity and stability of the rules and regulations of a country and availability of technology are enhancers of entrepreneurial spirit and activity of Romanian potential entrepreneurs. Thus, for new business ventures to reach their true potential, they need to be provided enhanced assistance in regards to clearly articulated and understandable legislation provisions [60], which make their access towards funding opportunities easier, and help them effectively exploit market opportunities to generate benefits for all interested parties. Many works out of the analyzed literature [48,59,60] draw attention to institutions and infrastructure either as catalysts or restrainers of entrepreneurial drive and activity. Furthermore, other studies [94] confirm that technology availability and accessibility support the entrepreneurial process by enhancing the potential of collaboration and collective intelligence to design and launch more robust and sustainable entrepreneurial projects.

Based on the data for Romania that were available to us and analyzed in the paper, we were, unfortunately, unable to prove whether a supportive education system, which equips potential entrepreneurs with the skills that they need for entrepreneurial activity, impacts the development of the entrepreneurial spirit. This contradicts the results of other works that found a direct influence of accumulated skills and knowledge through education on entrepreneurial activity [62,95,96]. Hence, this would be interesting for future investigation.

As a theoretical implication, the paper contributes to the entrepreneurship cognition literature by providing empirical evidence on the country's environment predicting factors that influence the development of its entrepreneurial spirit. The results of our research show that there are five prevalent factors in the country's entrepreneurial climate which predict its entrepreneurial drive and spirit: clear and stable operating environment, supportive education system, manageable taxes, technology availability and beneficial economic situation. Four (out of five) of our hypotheses were able to be

proven. Our research results suggest that, in transitional economies like Romania, clear and stable rules and regulations, manageable taxes, an overall beneficial economic situation, as well as availability and accessibility of technology, may lead to greater entrepreneurial drive and ambition, which are fundamental to sustainable economic growth.

From a practical perspective, the findings of the current research can help inform future policies aimed at stimulating national entrepreneurship in the transitional economies in Europe by delineating the most significant environmental factors impacting the entrepreneurial spirit of individual or groups of countries.

Our research has certain limits related to the nature and type of environment predicting factors that influence the development of entrepreneurial spirit. The limited number of factors is given by the specificity of the research tool used, a global structured survey developed by Amway, which collects specific data worldwide on certain entrepreneurship-related issues [16]. As such, we were not able to analyze the contextual factors or country-specific factors regarding the socio-economic development in Romania and this needs further attention in future research. Additionally, based on the data for Romania that were available to us, we were unable to prove whether the education system significantly affects the development of the entrepreneurial system. In order to further test this hypothesis in a meaningful way, a larger number of participants in Romania should be interviewed in future studies or complementary sources of data should be considered, like GEM data. Another limitation is represented by the measures of variables used, which were assessed by means of a single item [6]. However, previous studies have shown that single-item measures of well-defined constructs are reliable in individual- or country-level investigations [97].

As future avenues for research further theoretical and empirical research on individual, group or cross-countries' environment predictors of entrepreneurial spirit and potential are needed to help guide future research, as well as improve the consistency and relevance of national economic strategies aimed at supporting high-quality entrepreneurship. The use of up-to-date global entrepreneurship data can help further our understanding of the existing relationships between entrepreneurial spirit and its determinants in individual and groups of countries.

**Author Contributions:** This article is a contribution of two authors. C.P. conceived and developed the research idea, designed the methodology, analyzed the data and discussed the results. E.M. reviewed the literature and draw the conclusions. All authors have read and agreed to the published version of the manuscript.

**Funding:** This research received no external funding.

**Acknowledgments:** Authors are thankful to AMWAY for providing access to country's data reported in AGER 2018. They are also thankful to the editors and reviewers for their thoughtful comments, constructive criticisms and valuable suggestions, which made possible improvement and finalization of this article.

**Conflicts of Interest:** The authors declare no conflict of interest.

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
