# Peer review of "Country’s Entrepreneurial Environment Predictors for Starting a New Venture—Evidence for Romania"

_sustainability, doi:10.3390/su12187794_

Round 1
Reviewer 1 Report
Dear Authors,
Thanks for the opportunity to read your paper on the drivers of entrepreneurship in a transition economy. The topics discussed in this work are particularly interesting in the European context: however, a strong revision is necessary before publication, because there are several issues that still need to be addressed.
- The focus on a transition economy such as Romania must be discussed in detail. The brief justification provided in the Introduction is not enough.
- The studies cited in the Introduction for supporting the idea that entrepreneurship and self-employment are less attractive for individuals are out of topic and partially contradict the current literature on entrepreneurship: this issue is particularly complex and it is influenced by several factors (cultural, social, and economic), it cannot be reduced to a short overview as you did at page 1.
- This paper lacks in originality in its research objectives. As you point out at page 2, lines 60-62, this work is an update of a previous paper using a novel database with more recent data, but we are not talking about a difference of a 10 years period, we are talking about 1 year difference. Moreover, there are several empirical studies who have explored the influence of environmental factors such as fiscal system, education system, technology, etc..on entrepreneurship (see for example Colombo & Grilli (2005). Founders' human capital and the growth of new technology-based firms: A competence-based view. Research Policy 34(6)).
- Page 2, lines 67-68: it is not true that there is a lack of empirical country-level studies on the relation between entrepreneurial spirit and education. See Morakinyo & Akinsola (2019). Leadership and entrepreneurship education as a strategy for strengthening youth community engagement in Nigeria: Lessons learnt from jumpstart project, or Paz Marcano et al. (2020). Entrepreneurial profile in Venezuelan university education, among the most recent ones.
- The discussion about regional differences in Romania (especially in the Conclusions) is unrelated to the main issues discussed in the paper; this theme is not investigated through the model, neither highlighted in the literature review, therefore I suggest to avoid its discussion.
- My main concern about this paper is related to the empirical model. The linear regression model is very simple and its results are easy to understand. However, there are no control variables included in the model, such as age, gender, education level, etc.; there are no descriptive statistics of the dependent and independent variables (only mean and standard deviation in Table 2), and it is not clear which values they can assume based on the AGER questions. Finally, independent variables are created using the "perceptions" of the respondents; however, since you do not have a longitudinal database, it is hard to detect a causality effect. For example, the perception of the national fiscal system can be biased by the entrepreneurial spirit of the respondent. Is it possible to find more "objective" measures of operating environment, education system, taxes, technology, and economic situation?
- Check for typos in the text (example: page 8, line 293).
Author Response
Response to Reviewer 1 Comments
We would like to thank the reviewer for careful and thorough reading of this manuscript and for the thoughtful comments and suggestions, which were of great help in improving the manuscript. According to the suggestions, the revised manuscript has been systematically improved with new information (colored in red in the text of the paper). Our responses to the referee’s comments are given below.
We hope that the reviewer will find our responses satisfactory, and we are willing to finish the revised version of the manuscript including any further suggestion that the reviewer may have.
Please, find below the referee’s comments repeated and our responses in red inserted after each comment.
Looking forward to hearing from you soon.
Sincerely,
The Authors
Point 1: “The focus on a transition economy such as Romania must be discussed in detail. The brief justification provided in the Introduction is not enough.”
Response 1: The introduction has been significantly rewritten to indicate more clearly the challenges of entrepreneurship in a transition economy like Romania.
Point 2: “The studies cited in the Introduction for supporting the idea that entrepreneurship and self-employment are less attractive for individuals are out of topic and partially contradict the current literature on entrepreneurship: this issue is particularly complex and it is influenced by several factors (cultural, social, and economic), it cannot be reduced to a short overview as you did at page 1.”
Response 2: We thank the reviewer for the constructive comments offered on this matter. As per reviewer's comments, we added new discussion in the introduction to explain more clearly the conditions of the external entrepreneurial environment.
Point 3: “This paper lacks in originality in its research objectives. As you point out at page 2, lines 60-62, this work is an update of a previous paper using a novel database with more recent data, but we are not talking about a difference of a 10 years period, we are talking about 1 year difference. Moreover, there are several empirical studies who have explored the influence of environmental factors such as fiscal system, education system, technology, etc..on entrepreneurship (see for example Colombo & Grilli (2005). Founders' human capital and the growth of new technology-based firms: A competence-based view. Research Policy 34(6)).”
Response 3: While we can agree that the research objectives are not that ‘sharp’, given the challenges of the topic explored, we argue that the research is particularly relevant for policy makers who are involved in the development of strategies and policies meant to stimulate entrepreneurship in Romania. As per the reviewer’s comments, we have revised the objectives of the paper to improve clarity and enhance the flow of discussion (see lines 118-120 and 306-311). We would like to mention that while the previous study referred to in the paper discussed the individuals’ factors that drive the entrepreneurial desire in Romania (based on AGER 2017 data), the present work adds research evidence on country’s environmental factors that drive the entrepreneurial ambition in Romania (based on AGER 2018 data). The latest category of data didn’t make the purpose of previous AGER survey.
Point 4: “Page 2, lines 67-68: it is not true that there is a lack of empirical country-level studies on the relation between entrepreneurial spirit and education. See Morakinyo & Akinsola (2019). Leadership and entrepreneurship education as a strategy for strengthening youth community engagement in Nigeria: Lessons learnt from jumpstart project, or Paz Marcano et al. (2020). Entrepreneurial profile in Venezuelan university education, among the most recent ones.”
Response 4: We thank the reviewer for rigorousness on this matter. The paragraph was revised and corrected and new text added (see lines 142-148). We would like to mention that the literature which supports our research is extracted mainly from Web of Science. Other resources (except reports, statistics, etc.), which were not found in WOS, have been excluded.
Point 5: “The discussion about regional differences in Romania (especially in the Conclusions) is unrelated to the main issues discussed in the paper; this theme is not investigated through the model, neither highlighted in the literature review, therefore I suggest to avoid its discussion.”
Response 5: We agree. The text of the discussion part of the paper has been revised to comply better with the rest of the paper and the proposed hypotheses. We removed the irrelevant discussions and related references regarding regional differences in Romania.
Point 6: “My main concern about this paper is related to the empirical model. The linear regression model is very simple and its results are easy to understand. However, there are no control variables included in the model, such as age, gender, education level, etc.; there are no descriptive statistics of the dependent and independent variables (only mean and standard deviation in Table 2), and it is not clear which values they can assume based on the AGER questions. Finally, independent variables are created using the "perceptions" of the respondents; however, since you do not have a longitudinal database, it is hard to detect a causality effect. For example, the perception of the national fiscal system can be biased by the entrepreneurial spirit of the respondent. Is it possible to find more "objective" measures of operating environment, education system, taxes, technology, and economic situation?.”
Response 6: We thank the reviewer for these insights. The regression analysis was conducted again, using age, gender, education and working status as control variables, as per the reviewer’s suggestions (see section 4.2 Results). We added new text to explain the values taken by the variables based on the AGER questions and to support analysis of the results.
Point 7: “Check for typos in the text (example: page 8, line 293).”
Response 7: A Spelling and grammar check was run again.
Thanks to the reviewer’s thoughtful comments, constructive criticisms and valuable suggestions, the manuscript has been successfully improved and the revised version submitted.
Reviewer 2 Report
The introduction considers the importance of studying external factors that affect entrepreneurial intention. When considering the study in Romania, a brief reference on entrepreneurship and the macroeconomic environment in that country should be presented. Likewise, the objectives of the study should be identified more clearly.
In the literature review section, limitations imposed by government regulations and how their development can affect the development of entrepreneurship are indicated. More results from empirical studies that analyze the effect of institutional decisions on entrepreneurship should be indicated. Similarly, in the section on education, the role of incubators is discussed, but development is hardly specified. Other factors are indicated in the review. It should be explained in this review why only these factors are explained and results from other empirical studies should be presented.
In the methodology the author should better explain the information that AGER contains. They talk about the survey but something must be indicated regarding the significance of the sample. Likewise, the measurement scales of the analyzed variables should be explained. Likewise, what is the Entrepreneurial Spirit variable should be better defined.
The results section begins at 4.2. There is a duplication of tables or images that provide the same information. For example: only one of the two sources or table 1 or image 4. Same with table 6 or figure 5. Justify the use of ANOVA in the study. As results are available for the world average, the results of the model should be compared with those obtained at the global level, in order to present characteristic elements of the economy studied. The model being a linear regression could incorporate control variables that will reinforce the significance of the results.
Conclusions: Some of the paragraphs that appear in the conclusions would be more relevant in the introduction when presenting a context of the country analyzed. The conclusions presented are consistent, as are the limitations.
Author Response
Response to Reviewer 2 Comments
We would like to thank the reviewer for careful and thorough reading of this manuscript and for the thoughtful comments, constructive criticisms and valuable suggestions, which were of great help in improving the manuscript. According to the suggestions, the revised manuscript has been systematically corrected and improved with new information and additional interpretations (coloured in red in the text of the paper). Our responses to the referee’s comments are given below.
We hope that the reviewer will find our responses satisfactory, and we are willing to finish the revised version of the manuscript including any further suggestion that the reviewer may have.
Please, find below the referee’s comments repeated and our responses in red inserted after each comment.
Looking forward to hearing from you soon.
Sincerely,
The Authors
Point 1: “The introduction considers the importance of studying external factors that affect entrepreneurial intention. When considering the study in Romania, a brief reference on entrepreneurship and the macroeconomic environment in that country should be presented. Likewise, the objectives of the study should be identified more clearly.”
Response 1: We thank the reviewer for the constructive comments offered on this matter. The introduction has been significantly rewritten to indicate more clearly the challenges of entrepreneurship and of macroeconomic environment in Romania. As per the reviewer’s comments, we have also revised the objectives of the paper to improve clarity and enhance the flow of discussion (see lines 118-120 and 306-311).
Point 2: “In the literature review section, limitations imposed by government regulations and how their development can affect the development of entrepreneurship are indicated. More results from empirical studies that analyze the effect of institutional decisions on entrepreneurship should be indicated. Similarly, in the section on education, the role of incubators is discussed, but development is hardly specified. Other factors are indicated in the review. It should be explained in this review why only these factors are explained and results from other empirical studies should be presented.”
Response 2: We have added new text to address these issues in the Introduction and Literature review sections. Research limitations regarding the selection of entrepreneurial environment factors are presented are the end. We would like to mention that the research relies on data collected by Amway on questionnaire-based survey. The questionnaire is fully structured and the questions are addressed in a given order and following a clear structure. Our research relies on AGER data 2018 and AGER reports, and uses only the information collected on the basis of this global survey. We didn’t aim at this point to design a more extended research and add new data (i.e. regarding other contextual factors specific to entrepreneurship in Romania) as existing data did offer interesting perspectives on perception about entrepreneurship and entrepreneurial spirit. Yet, further directions for future research are included in the conclusion section.
Point 3: “In the methodology the author should better explain the information that AGER contains. They talk about the survey but something must be indicated regarding the significance of the sample. Likewise, the measurement scales of the analyzed variables should be explained. Likewise, what is the Entrepreneurial Spirit variable should be better defined.”
Response 3: As per reviewer's comments, we have added in the Methodology new discussion about AGER tool, sample (see lines 312-322), and measurement scales. The, entrepreneurial spirit variable is defined in lines 182-187 and 338-339.
Point 4: “The results section begins at 4.2. There is a duplication of tables or images that provide the same information. For example: only one of the two sources or table 1 or image 4. Same with table 6 or figure 5. Justify the use of ANOVA in the study. As results are available for the world average, the results of the model should be compared with those obtained at the global level, in order to present characteristic elements of the economy studied. The model being a linear regression could incorporate control variables that will reinforce the significance of the results.”
Response 4: We appreciate the rigorousness of the reviewer on this matter. We have revised the text and removed several figures mentioned by the reviewer and a table to avoid duplication.
The regression analysis was conducted again, using age, gender, education and working status as control variables, as per the reviewer’s suggestions (see section 4.2 Results). The use of regression model in the study was explained (see lines 342-351).
Point 5: “Conclusions: Some of the paragraphs that appear in the conclusions would be more relevant in the introduction when presenting a context of the country analyzed. The conclusions presented are consistent, as are the limitations.”
Response 5: We agree. The text of the discussion part of the paper has been revised to comply better with the rest of the paper and the proposed hypotheses. We removed the irrelevant discussions and related references, and added relevant text in the introduction, as per reviewer’s suggestions.
Thanks to the reviewer’s thoughtful comments, constructive criticisms and valuable suggestions, the manuscript has been successfully improved and the revised version submitted.
Reviewer 3 Report
Statistical analysis is good. But research question and formwork is not sharp. Then objective is not clear.
For example, is Money of figure 4 relate with Tax of figure 5?
Matching between theoretical concept and analyzed results is necessary for readers to understand authors' intent or emphasis.
Author Response
Response to Reviewer 3 Comments
We would like to thank the reviewer for careful and thorough reading of this manuscript and for the thoughtful comments, constructive criticisms and valuable suggestions, which were of great help in improving the manuscript. According to the suggestions, the revised manuscript has been systematically corrected and improved with new information and additional interpretations (coloured in red in the text of the paper). Our responses to the referee’s comments are given below.
We hope that the reviewer will find our responses satisfactory, and we are willing to finish the revised version of the manuscript including any further suggestion that the reviewer may have.
Please, find below the referee’s comments repeated and our responses in red inserted after each comment.
Looking forward to hearing from you soon.
Sincerely,
The Authors
Point 1: “Statistical analysis is good. But research question and formwork is not sharp. Then objective is not clear. For example, is Money of figure 4 relate with Tax of figure 5?”
Response 1: We appreciate the constructive feedback from the reviewer on this matter. While we can agree that the research objectives are not ‘sharp’ in the given situation, we argue that the research is particularly relevant for supporting policy makers with development process of strategies and policies that stimulate entrepreneurship in Romania. As per the reviewer’s comments, we have revised the objectives of the paper to improve clarity and enhance the flow of discussion (see lines 118-120 and 306-311), as well as the results (see 4.2).
Point 2: “Matching between theoretical concept and analyzed results is necessary for readers to understand authors' intent or emphasis.”
Response 2: We thank the reviewer for these insights. The relation between theoretical concept and the results has been checked and enhanced, and the corresponding text revised. Some irrelevant figures, tables or paragraphs have been removed and new text was added.
Thanks to the reviewer’s constructive criticisms, the manuscript has been successfully improved and the revised version submitted.
Round 2
Reviewer 1 Report
The authors have addressed my requests, I am satisfied with the revision
Reviewer 2 Report
The authors have substantially modified the introduction, focusing and justifying the research topic in a more clear and concise manner. Likewise, the source of data used appears better defined. The results part is presented in a clearer way.
Reviewer 3 Report
Quality was improved. Education system needs to be tested.